# TNASP: A Transformer-based NAS Predictor with a Self-evolution Framework

**Shun Lu[1,2], Jixiang Li[3], Jianchao Tan[3], Sen Yang[3], Ji Liu[3]**
[1] Research Center for Intelligent Computing Systems, State Key Laboratory of Computer Architecture,
Institute of Computing Technology, Chinese Academy of Sciences
[2] University of Chinese Academy of Sciences
[3] Kuaishou Technology
`lushun19s@ict.ac.cn, {lijixiang,jianchaotan,senyang,jiliu}@kuaishou.com`

## Abstract

Predictor-based Neural Architecture Search (NAS) continues to be an important topic because it aims to mitigate the time-consuming search procedure of traditional NAS methods. A promising performance predictor determines the quality of final searched models in predictor-based NAS methods. Most existing predictor-based methodologies train model-based predictors under a proxy dataset setting, which may suffer from the accuracy decline and the generalization problem, mainly due to their poor abilities to represent spatial topology information of the graph structure data. Besides the poor encoding for spatial topology information, these works did not take advantage of the temporal information such as historical evaluations during training. Thus, we propose a Transformer-based NAS performance predictor, associated with a Laplacian matrix based positional encoding strategy, which better represents topology information and achieves better performance than previous state-of-the-art methods on NAS-Bench-101, NAS-Bench-201, and DARTS search space. Furthermore, we also propose a self-evolution framework that can fully utilize temporal information as guidance. This framework iteratively involves previous evaluation information as constraints into current optimization iteration, thus further improving the performance of our predictor. Such framework is model-agnostic, thus can enhance performance on various backbone structures for the prediction task. Our proposed method helped us rank 2nd among all teams in CVPR 2021 NAS Competition Track 2: Performance Prediction Track.

## 1 Introduction

Neural Architecture Search (NAS) aims to automatically find out superb architectures in a pre-defined search space. The NAS models have outperformed human-designed models in many domains [37, 18, 13, 3, 15]. However, traditional NAS methods like Reinforcement learning and Evolutionary learning require enormous computation resources, i.e., hundreds or thousands of GPU hours, to train sub-models to obtain their performance estimation. The intensive computation prohibits NAS models from deployments to real applications. Although differentiable NAS methods [26] have less search time than traditional methods, they usually suffer from performance collapse due to several problems such as optimization gap [6], discretization discrepancy and the unfair advantage [9], trapping into sharp minimum and the dominant eigenvalue [49, 5].

To reduce the search cost in NAS, predictor-based NAS methods [30, 23, 43, 29, 32, 7, 46, 4, 1] use performance predictors to predict the accuracy of architectures quickly instead of training all architectures to get the accuracy. Simple training-free predictors have been shown promising in some applications, however, those performances are usually not good enough in practice. Many

35th Conference on Neural Information Processing Systems (NeurIPS 2021).

works focus on how to design effective training-based predictors [43, 29, 32, 7, 46]. A training-based predictor usually consists of an encoder module and a regressor module, and only needs to learn from a few architecture-accuracy sampled pairs, thus leading to a fast learning procedure. After the performance predictor learns the relationship between the architecture and its corresponding accuracy, it is able to predict the performance of other unseen architectures in the search space, which greatly accelerates the search process of NAS.

One of the key components of a performance predictor is to encode discrete architectures into continuous feature representations. Neural predictor[43] and CTNAS[7] applied GCN [20] to capture the feature representation of input model structures. Both SemiNAS [29] and GATES [32] learned an embedding matrix for the candidate operations in the search space, and represented architectures as a combination of different embeddings. ReNAS[46] calculated the types matrix, flops matrix, and parameters matrix, and then concatenated them together to form a feature tensor to represent a specific architecture. Unlike previous methods, we propose a **T**ransformer-based **NAS** performance **P**redictor (**TNASP**) and use the linear transformation output over the Laplacian matrix of the model structure graph to be the positional encoding.

There are several advantages of the Transformer that can be used to train a good performance predictor. First, the self-attention module can help explore better feature representations from the graph structure data. Second, the multi-head mechanism can further help encode the different subspace information at different positions from the graph structure data, as also claimed by the original Transformer paper. Third, the Laplacian matrix based positional encoding method also fits well to find topology position information on the graph. In summary, we demonstrate that Transformer is an effective method to extract feature representation from discrete architecture graphs, and also has superb generalization abilities for processing unseen data, as shown in our experiments. Since the predictors are trained on a small size proxy dataset but the test dataset is much larger, training-based NAS predictors usually have poor generalization abilities. The powerful Transformer encoder can mitigate this problem to a certain degree, due to its good ability to encode topology information.

To further improve our predictor, we introduce a self-evolution framework, which makes full use of temporal information to guide the training. The framework iteratively involves each evaluation score of the previously predicted results on a validation dataset into a gradient based optimization iteration as constraints, to push the predictions close to ground truths gradually. We demonstrate the framework can make our predictor have better generalization, and also achieve better performance than previous predictors.

Our proposed framework is fruitful and practical in several scenarios, for example, all of the machine learning competitions. In competition, the test set that the host provided to users is exactly the validation role in our setting. And the temporal information, such as the feedback of each submission, is able to gradually improve original training iteratively with our framework. Generally speaking, our contributions can be summarized as follows:

- We propose a Transformer-based NAS performance predictor (TNASP) to better encode the spatial topology information, utilizing the multi-head self-attention mechanism to map the discrete architectures to a meaningful feature representation and applying the linear transformation of the Laplacian matrix as the positional encoding.

- By leveraging each evaluation score information in history as constraints in training, and applying a gradient based optimization method to iteratively solve the constrained optimization problem, we introduce a generic self-evolution framework to further improve the performance of the proposed predictor, making full use of temporal information.

- Our proposed method surpasses the previous state-of-the-art methods under the same proxy training dataset and achieves state-of-the-art results on NAS-Bench-101 [48], NAS-Bench-201[14], and DARTS [26] search space.

## 2   Related Work

Due to the enormous search cost of traditional NAS methods such as reinforcement learning based methods [52, 2] and evolutionary methods [34, 27], network performance predictor based NAS methods have become a pretty active topic recently. Most works only require a few architecture-accuracy data pairs to train a predictor and estimate the performance of unseen architectures, which

can be categorized as training-based network performance predictors. Other works propose to calculate some kinds of metrics over network structure to represent network performance without training, which are denoted as training-free network performance predictors.

**Training-based network performance predictors** Training-based predictors follow a unified paradigm to learn the correlation from network architectures and their corresponding accuracy. As it's hard to learn useful features directly from the discrete network architectures, various methods have been explored to map the discrete representation into a continuous latent space and can be roughly divided into sequence-based schemes and graph-based methods. Sequence-based schemes denote each architecture by a discrete sequence with fixed length and use MLP [25, 44, 46], embedding matrix [11, 32], Auto-Encoder [30, 51, 29] or GBDT [28] to convert the sequence into continuous representation. However, graph-based methods treat the architecture as a graph and use the graph-format data i.e. adjacency matrix and node features as inputs. Diverse graph processing methods such as GHN [50], GCN [22, 43, 36, 7], GIN [47] and WL-Kernel [35] have been tried. Similar to these methods, our Transformer-based predictor also belongs to the training-based network performance predictors category. Differently, we propose a novel encoding scheme by utilizing the Laplacian matrix as the positional encoding and leveraging the multi-head self-attention mechanism from the Transformer [41] to encode network architectures to excavate more representative features.

**Training-free network performance predictors** Recently, several works proposed to predict the network performance by designing some metrics, without training a model to perform prediction. [31] scored networks at initialization by computing correspondence between binary codes in the whole mini-batch. TE-NAS [4] analyzed trainability and expressivity to evaluate each network performance. Zero-Cost NAS [1] compared six proxies to predict network performance, such as grasp [42], fisher [39, 40], synflow [38], and so on, but failed to rank architectures in NAS-Bench-101 [48]. Zen-NAS [24] measured the expressivity of a network by expected Gaussian complexity to represent its performance. Although these training-free network performance predictors are efficient and have achieved satisfying results on several datasets, their robustness can not be guaranteed and the performance fluctuated dramatically among different tasks. Compared to these methods, our training-based predictor is more time-consuming, but its performance is dramatically better.

## 3 Methods

In this section, we first briefly review the common paradigm for the training-based NAS predictors as preliminary. Following the same paradigm, we then introduce the details of our Transformer-based predictor. Furthermore, we introduce a general framework to utilize the information in evaluation histories over a validation dataset as constraints in a gradient based optimization to advance the predictor by a self-evolution framework.

### 3.1 Training-based network performance predictors

Previous works [30, 43, 46, 7] proposed to first apply an encoder $f_E$ to encode the discrete architectures into continuous feature representations, which can be formulated as:

$$e = f_E(A, \kappa) \tag{1}$$

where $A \in \mathbb{R}^{N \times N}$ denotes the adjacency matrix and stands for the directed acyclic connections between nodes, $N$ denotes the number of the nodes. $\kappa \in \mathbb{R}^{N \times F}$ stands for the feature matrix and represents the characteristics of the nodes, $F$ denotes the output dimension of the embedding extractor. For NAS predictors, the adjacency matrix $A$ shows the topology information of an architecture and $\kappa$ usually indicates the representation of operations for nodes or edges. Among previous works, the encoder $f_E$ can be a GCN or LSTM or simply an embedding matrix, and the embedding vector $e$ can be interpreted as a latent representation of a specific architecture.

After encoding the discrete architecture into a continuous representation using an encoder, it is easier and more accurate to estimate the network accuracy $\hat{y}$ by a simple regressor $f_R$ once the embedding vector $e$ meaningfully represents the architecture in the latent space.

$$\hat{y} = f_R(e) \tag{2}$$

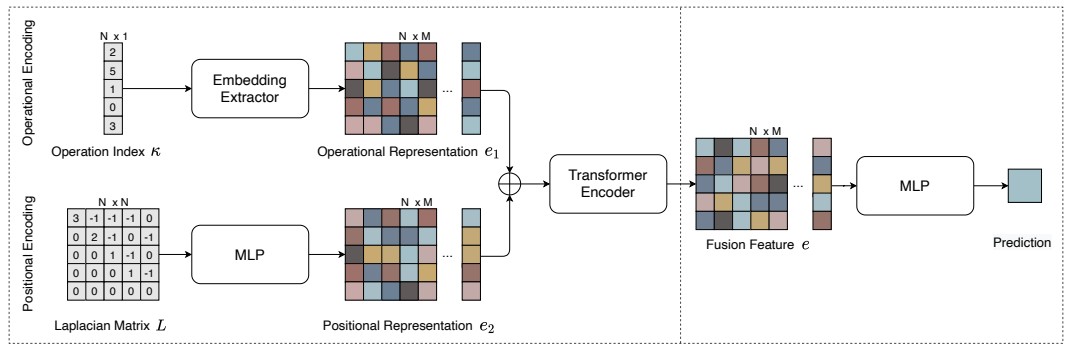

Figure 1: Our Transformer-based NAS predictor mainly consists of an encoder and a regressor. We first encode the information of operations and connections into continuous representation, followed by 3 Transformer encoder layers, and the regressor uses the output feature of Transformer encoder layers to derive the final prediction. Color grey in features denotes integral form and other colors denote float form. $N$, $F$ and $M$ denote the number of the nodes, the number of candidate operations for each node and the latent dimension of the continuous feature, respectively.

Generally, we apply the MSE loss to supervise the training of the encoder and the regressor, and we use $\theta$ to denote the parameters of both encoder $f_E$ and regressor $f_R$. The optimal parameters $\theta^\star$ can be optimized by stochastic gradient descent method:

$$\theta^\star = \arg\min_\theta \frac{1}{n} \sum_{i=1}^{n} (\hat{y}_i - y_i)^2 \tag{3}$$

where $n$ represents the total number of the training data, $y_i$ denotes the ground truth of the $i$-th sample.

### 3.2 Transformer-based predictor

Recently, the Transformer architecture [41] has attracted much attention and outperformed many state-of-the-art models due to its powerful representation ability, which motivates us to borrow the decent encoder structure to be our predictor backbone.

As shown in Fig.1, we first get the operation feature $e_1 \in \mathbb{R}^{N \times M}$ by transforming the operation vector $\kappa$ with an embedding matrix $E \in \mathbb{R}^{F \times M}$.

$$e_1 = E(\kappa) \tag{4}$$

To characterize the positional information, we have tried various encoding methods, and find that the Laplacian matrix is an effective scheme and has the potential to better represent topology information of graph structure data as it contains both the connectivity of a graph and the centrality of each node. Thus we propose to encode the positional information by utilizing the Laplacian matrix ($L \in \mathbb{R}^{N \times N}$), computed from the adjacency matrix ($A \in \mathbb{R}^{N \times N}$) and the degree matrix ($D \in \mathbb{R}^{N \times N}$):

$$L = D - A \tag{5}$$

We did not choose the normalized Laplacian matrix, because we find it does not perform well in our experiments.

Then we use a linear layer to map the the Laplacian matrix ($L$) to a continuous feature vector $e_2$, whose dimension is same as the embedding vector $e_1$:

$$e_2 = \text{MLP}(L) \tag{6}$$

Finally, we get the the continuous representation from the Transformer encoder with multi-head self-attention module:

$$e = E_{\text{Transformer}}(e_1 + e_2) \tag{7}$$

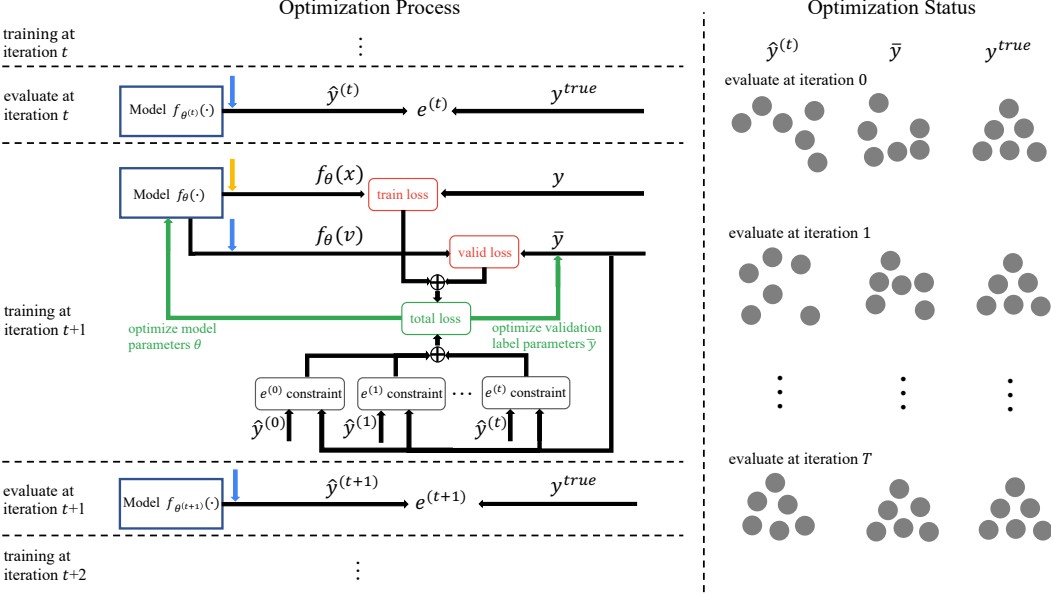

Figure 2: The left sub-figure illustrates the optimization process of self-evolution framework, the blue arrow denotes the input of the validation dataset and the yellow arrow stands for the input of the training dataset. The right sub-figure shows the status transition of the validation predictions and the trainable validation label parameters, both are approaching to ground truth gradually. All the symbols can be found in equations in Section 3.3.

As the continuous representation $e$ contains richer information about the architecture (including the operations and the connections) in the latent space, it's easier to learn the relationship between this representation and its corresponding accuracy. Therefore, we only choose a simple regressor, specifically 2 Multi-Layer Perceptions (MLP), to estimate the final accuracy. The experiments demonstrate the promising performance of our whole designs.

### 3.3 Self-evolution framework

Besides our predictor design that has good encoding ability for spatial topology information, we also propose a framework that can further improve performance by fully utilizing temporal information during training. This framework iteratively involves the evaluation information in the previous iteration over a validation dataset into the current iteration of optimization as constraints, as shown in Fig. 2. Our training strategy is useful and practical in several scenarios, for example, all of the machine learning competitions. In competition, the test set that the host provided to users is exactly the validation role in our framework. And the system feedback of each submission can gradually improve the original training iteratively by adding more and more historical evaluations as constraints. Specifically, our objective function can be formulated as below:

$$\min_{\theta, \bar{y}} \sum_{i=1}^{n} \|f_\theta(x_i) - y_i\|^2 + \alpha \sum_{j=1}^{V} \|f_\theta(v_j) - \bar{y}_j\|^2$$

$$\text{s.t.} \quad \frac{1}{V} \sum_{j=1}^{V} \|\hat{y}_j^{(t)} - \bar{y}_j\|^2 = e^{(t)}, t = 1, 2, 3, ..., T$$

(8)

where, $x$ is training data with size $n$. And we call $v$ as help data or validation data with size $V$ in this paper. We denote the number of evaluation on validation dataset $v$ with $t$. And $\alpha$ is the weight to balance train loss and validation loss. We choose different value for $\alpha$ in different experiments. $\bar{y}_j$ is an auxiliary variable that serves as the proxy of ground truth accuracy in validation dataset. $f_\theta(v_j)$ is current forward pass inference results of predictor $f_\theta(x)$ over validation data $v_j$, which is used in

back-propagation step to update model parameters $\theta$ at current iteration. $\hat{y}_j^{(t)}$ is the corresponding previous predictions at iteration $t = 1, 2, 3, ..., T$.

For $e^{(t)}$, it should be computed in the system back-end, for example, in the competition situation, users can only get the final evaluation score instead of the ground truth label of every validation sample. For simplicity, we choose MSE as the evaluation metric between the historical prediction results $\hat{y}_j^{(t)}$ and the user-unknown ground truth $y_j^{true}$. In a practical situation, the metric can be more complicated. $y_j^{true}$ is user-unknown, which has diverse meanings in different situations. On the one hand, for the Machine Learning competitions, at $t_{th}$ submission, we submit our predictions $\hat{y}_j^{(t)}$ to the system, and the system will compute the $e^{(t)}$ at the back-end privately and return only $e^{(t)}$ to us, and thus we call $y_j^{true}$ is user-unknown. Our self-evolution system can make use of only these $e^{(t)}$ for $t = 1, 2, 3, ..., n$ to incrementally improve our predictor. On the other hand, for offline benchmark experiments conducted in this paper, we use evaluation error on validation set V in each previous iteration as $e^{(t)}$ to guide the current iteration training using our SE framework. We did know the ground truth on this validation set V, but we did not make use of this ground truth directly in the SE framework. We just mimic the aforementioned system of the Machine Learning competitions to compute $e^{(t)}$ as in equation 9, which is used in our evolving procedure. Thus the ground truth here is also called user-unknown.

$$e^{(t)} = \frac{1}{V} \sum_{j=1}^{V} \|\hat{y}_j^{(t)} - y_j^{true}\|^2, t = 1, 2, 3, ..., T \tag{9}$$

We formulate above Eq. 8 as a Minimax optimization problem using Lagrangian Multiplier:

$$
\begin{aligned}
L(\theta, \bar{y}, \lambda) = \min_{\theta, \bar{y}} \max_{\lambda} & \sum_{i=1}^{n} \|f_\theta(x_i) - y_i\|^2 + \alpha \sum_{j=1}^{V} \|f_\theta(v_j) - \bar{y}_j\|^2 \\
& + \frac{1}{T} \sum_{t=1}^{T} \lambda^{(t)} \left( \frac{1}{V} \sum_{j=1}^{V} \|\hat{y}_j^{(t)} - \bar{y}_j\|^2 - e^{(t)} \right)
\end{aligned}
\tag{10}
$$

At each training step, we adopt a gradient based optimization method to update the variables. And the whole framework is summarized in algorithm 1.

$$\theta^{k+1} = \theta^k - \eta_\theta \frac{\partial L(\theta, \bar{y}^k, \lambda^k)}{\partial \theta} \tag{11}$$

$$\bar{y}^{k+1} = \bar{y}^k - \eta_{\bar{y}} \frac{\partial L(\theta^k, \bar{y}, \lambda^k)}{\partial \bar{y}} \tag{12}$$

$$\lambda^{k+1} = \lambda^k + \eta_\lambda \frac{\partial L(\theta^k, \bar{y}^k, \lambda)}{\partial \lambda} \tag{13}$$

In summary, the self-evolution framework can make full use of any available information (either the competition system historical submission feedback or historical evaluation information on validation dataset during model training) to guide the predictor training to avoid over-fitting, thus generalized well on the test dataset. Moreover, this framework directly treats each historical validation evaluation information as each hard constraint during training and reformulates the whole constrained training problem as a minimax optimization problem, solved by a gradient-based optimization method efficiently and effectively.

## 4 Experiments

We employ our TNASP on three different search spaces, specifically NAS-Bench-101[48], NAS-Bench-201[14] and DARTS[26]. Moreover, we put several experimental results performed on ImageNet [21]; a comprehensive comparison with GCN, SemiNAS [29], BONAS [36]; the implementation details; searched architecture visualizations in supplementary materials.

**Algorithm 1** Self-evolution Optimization Algorithm

---

**Input:** Input training data $x$, input validation data $v$, input training target $y$, neural network $f$.
**Output:** Network parameters $\theta$, estimated target $\bar{y}$.

1: Optimize the network parameters $\theta$ until convergence using normal training performed on the training dataset only.
2: **for** $t = 1$ **to** $T$ **do**
3:     Compute $e^{(t)}$ according to Eq. (9) using predictor's prediction results on validation dataset.
4:     Add a new constraint: $\frac{1}{V} \sum_{j=1}^{V} \|\hat{y}_j^{(t)} - \bar{y}_j\|^2 = e^{(t)}$ into Eq. (8)
5:     **while** not converged **do**
6:         Update $\theta$ according to the Eq. (11)
7:         Update $\bar{y}$ according to the Eq. (12)
8:         Update $\lambda$ according to the Eq. (13)
9:     **end while**
10: **end for**
11: **return** Network parameters $\theta$ and estimated targets $\bar{y}$

---

## 4.1 Experiments on NAS-Bench-101

**NAS-Bench-101** There are 423,624 different architectures in NAS-Bench-101[48]. Each architecture is stacked by 9 repeated cells. The maximum number of nodes and edges for each cell are 7 and 9, respectively. Nodes represent different candidate operations and edges show the connection between nodes. NAS-Bench-101[48] provides the validation accuracy and the test accuracy for three different runs. Following Neural Predictor [43], we utilize the validation accuracy from a single run as the training target and apply the mean test accuracy over three runs as the ground truth accuracy for evaluating the performance of our predictions. We train all models for 300 epochs with batch size 10 using Adam optimizer and the learning rate is 1e-4 with a cosine decay strategy.

**Comparison with SOTA methods** Following the settings in [43, 7, 46], we choose 0.02%, 0.04%, 0.1% and 1% of the whole data as our training set to train our predictor. And we use all the data as a test set to calculate Kendall's Tau to evaluate the performance of different predictors. The results are shown in Tab.1. We can see that when the training data is extremely deficient (only 0.02% and 0.04%), our predictor achieves obviously higher Kendall's Tau than Neural Predictor [43] and NAO [30], which illustrates the stronger representation capability of our method in few-shot training scenarios. When using 0.1% of the whole data as the training set, our method still outperforms theirs. When the training data size becomes larger(1%), the performance of all predictors has been improved obviously, due to more information gained. However, our method still beats them.

Although our TNASP has got the best results in all kinds of data splits, we demonstrate that the performance can be further improved with our proposed self-evolution(SE) framework. We use another 200 data as the validation set and only apply the MSE loss over these validation data as the constraints to guide the training optimization of our predictor. As shown in Tab.1, when applied with our SE framework, TNASP can get higher Kendall's Tau under a variety of different data splits. Furthermore, our framework is generic and easy to combine with other methods. When applied to Neural Predictor [43] and NAO [30], both methods achieve higher Kendall's Tau as shown in Tab.1, fully demonstrating the effectiveness of our proposed self-evolution framework.

## 4.2 Experiments on NAS-Bench-201

**NAS-Bench-201** Architectures in NAS-Bench-201 [14] are constructed with repeated cells. All cells are composed of 4 nodes and 6 edges, and every cell inside an architecture shares the same structure. Each edge represents an operation, and there are 5 candidate operations in total, resulting in 15,625 different cell candidates. NAS-Bench-201 [14] provides three different results of each architecture on three different datasets and we choose CIFAR-10 results as our targets.

**Comparison with SOTA methods** Similar to the experimental setup on NAS-Bench-101 [48], we directly train predictors ranging from 0.05% to 10% of the whole dataset and also evaluate the performance on the whole dataset. As shown in Tab.2, TNASP consistently achieves the highest

| Training Samples | 100 (0.02%) | 172 (0.04%) | 424 (0.1%) | 424 (0.1%) | 4236 (1%) |
|---|---|---|---|---|---|
| **Validation Samples** | 200 | 200 | 200 | 200 | 200 |
| **Test Samples** | all | all | 100 | all | all |
| Neural Predictor[†] [43] | 0.391 | 0.545 | 0.710 | 0.679 | 0.769 |
| SPOS [16] | - | - | 0.196* | - | - |
| FairNAS [8] | - | - | -0.232* | - | - |
| NAO[‡] [30] | 0.501 | 0.566 | 0.704 | 0.666 | 0.775 |
| ReNAS [46] | - | - | 0.634* | 0.657 | 0.816 |
| RegressionNAS | - | - | 0.430* | - | - |
| CTNAS [7] | - | - | 0.751* | - | - |
| TNASP | **0.600** | **0.669** | **0.752** | **0.705** | **0.820** |
| Neural Predictor[†] + SE | 0.458 | 0.577 | 0.713 | 0.684 | 0.773 |
| NAO[‡] + SE | 0.564 | 0.624 | 0.732 | 0.680 | 0.787 |
| TNASP + SE | **0.613** | **0.671** | **0.754** | **0.722** | **0.820** |

Table 1: Comparison with other methods on NAS-Bench-101. We calculate the Kendall's Tau by predicting accuracy of all architectures in NAS-Bench-101. [†]: re-implemented by ourselves. [‡]: implemented based on their released model. *: reported by CTNAS[7].

| Training Samples | 78(0.05%) | 156(1%) | 469(3%) | 781(5%) | 1563(10%) |
|---|---|---|---|---|---|
| **Validation Samples** | 200 | 200 | 200 | 200 | 200 |
| **Test Samples** | all | all | all | all | all |
| Neural Predictor[†] [43] | 0.343 | 0.413 | 0.584 | 0.634 | 0.646 |
| NAO[‡] [16] | 0.467 | 0.493 | 0.470 | 0.522 | 0.526 |
| TNASP | **0.539** | **0.589** | **0.640** | **0.689** | **0.724** |
| Neural Predictor[†] + SE | 0.377 | 0.433 | 0.602 | 0.652 | 0.649 |
| NAO[‡] + SE | 0.511 | 0.511 | 0.514 | 0.529 | 0.528 |
| TNASP + SE | **0.565** | **0.594** | **0.642** | **0.690** | **0.726** |

Table 2: Comparison with other methods on NAS-Bench-201. We calculate the Kendall's Tau by predicting the accuracy of all architectures in NAS-Bench-201 and comparing them with ground truths. [†]: re-implemented by ourselves. [‡]: implemented based on their released model.

Kendall's Tau and has a large improvement compared to other models regardless of the amount of training data. When applied with the self-evolution framework, TNASP+SE obtains further higher Kendall's Tau at each proportion of the data. Neural Predictor [43] and NAO [30] have the maximum Kendall's Tau improvement of 0.034 and 0.044, respectively, indicating the self-evolution framework is effective for various predictors and different search spaces.

From the above observations, we can conclude that when the number of validation samples is greater than training samples, the predictor can get a larger performance improvement by using the SE-framework. On the contrary, the predictor only got little performance gains when the number of training samples is much larger than validation samples. This phenomenon happens not only on our Transformer-based predictor but also on Neural Predictor and NAO.

## 4.3 Experiments on DARTS

**DARTS** Architectures in DARTS search space are built by normal cells and reductions cells. Each cell consists of 7 nodes and 14 edges. Every edge is a selection from the 7 candidate operations (we omit the Zero operation in our experiments). Following CTNAS [7], we first train a supernet on CIFAR-10 with a uniform sampling strategy as in [16] for 120 epochs, to get proxy labels for architectures. After the supernet training, we randomly sample 1000 architectures in this search space and query their test accuracy by inheriting the supernet weights to construct the architecture-accuracy pair. Finally, we utilize the generated data pairs to train our TNASP model and apply it in an evolutionary algorithm [10] to search for good architectures in the search space.

| Architecture | Test Accuracy(%) | #Params.(M) | Search Cost(G·D) |
|---|---|---|---|
| DenseNet-BC [19] | 96.54 | 25.6 | - |
| PyramidNet-BC [17] | 96.69 | 26.0 | - |
| Random search baseline | 96.71 ± 0.15 | 3.2 | - |
| NASNet-A [53] + cutout | 97.35 | 3.3 | 1,800 |
| NASNet-B [53] + cutout | 96.27 | **2.6** | 1,800 |
| NASNet-C [53] + cutout | 96.41 | 3.1 | 1,800 |
| AmoebaNet-A [34] + cutout | 96.66 ± 0.06 | 3.2 | 3,150 |
| SNAS [45] | 97.02 | 2.9 | 1.5 |
| ENAS [33] + cutout | 97.11 | 4.6 | 0.5 |
| DARTS [26] + cutout | 97.24 ± 0.09 | 3.4 | 4 |
| NAONet [30] | 97.02 | 28.6 | 200 |
| PNAS [25] + cutout | 97.17 ± 0.07 | 3.2 | - |
| GHN [50] + cutout | 97.16 ± 0.07 | 5.7 | 0.8 |
| D-VAE [51] | 94.80 | - | - |
| NGE [22] + cutout | 97.40 | - | **0.1** |
| BONAS-A [36] + cutout | 97.31 | 3.45 | 2.5 |
| CTNAS [7] + cutout | 97.41 ± 0.04 | 3.6 | 0.3 |
| TNASP + cutout(avg) | **97.43 ± 0.04** | 3.6 ± 0.1 | 0.3 |
| TNASP + cutout(best) | **97.48** | 3.7 | 0.3 |

Table 3: Comparison with other methods in DARTS [26] search space on CIFAR-10. "cutout": evaluate the searched cells using cutout [12] data augmentation. "G·D": GPU days.

**Comparison with SOTA methods**    To avoid randomness, we select the top-3 searched architectures to re-train and report the average metrics in the Tab.3. We can notice that our searched cells achieve higher average test accuracy than all the other methods, and the best searched cells get the highest test accuracy 97.48, which implies that our predictor learns the effective correspondence between architectures and accuracy. We visualize the best searched architecture in Fig.3 and append the other searched cells in our supplementary material. Noticeably, several convolution operations appear in the reduction cell and the normal cell contains $max\_pool\_3 \times 3$, which rarely appears in other methods searched normal cells. Our searched cells have a novel structure and achieve higher test accuracy, showing that our method obtains a better local minimum than other methods. It's interesting to explore the essential differences between the cells of different methods in the future.

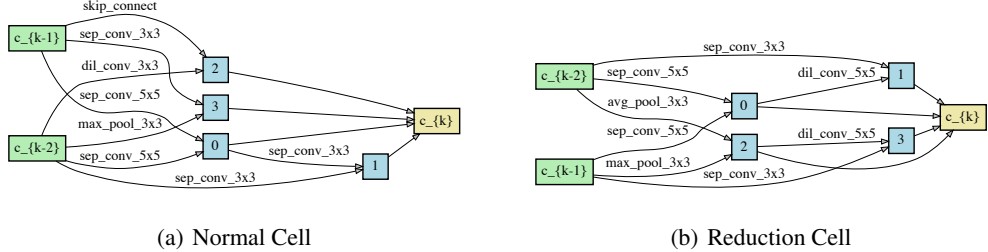

(a) Normal Cell                                   (b) Reduction Cell

Figure 3: Our best searched normal cell and reduction cell.

## 4.4   Ablation studies

**Different positional encoding strategies**    We compare different positional encoding schemes in Tab.4 on NAS-Bench-101 [48]. Adjacency option gets the highest score when training data size is 100, Laplacian option obtains highest Kendall's Tau in all other cases. Thus we choose the Laplacian matrix as the positional encoding in our experiments. Interestingly, the Normalized Laplacian option's performance is worse than the Laplacian option.

| Training Samples
Test Samples | 100 (0.02%)
all | 172 (0.04%)
all | 424 (0.1%)
100 | 424 (0.1%)
all | 4236 (1%)
all |
|---|---|---|---|---|---|
| Random | 0.374 | 0.366 | 0.469 | 0.362 | 0.365 |
| WL-PE | 0.439 | 0.483 | 0.509 | 0.519 | 0.628 |
| Normalized Adjacency | 0.549 | 0.645 | 0.685 | 0.671 | 0.786 |
| Adjacency | **0.620** | 0.643 | 0.725 | 0.693 | 0.802 |
| Normalized Laplacian | 0.592 | 0.642 | 0.740 | 0.697 | 0.813 |
| Laplacian Eigen-vector | 0.414 | 0.421 | 0.614 | 0.524 | 0.656 |
| Laplacian | 0.600 | **0.669** | **0.752** | **0.705** | **0.820** |

Table 4: Comparison with different positional encoding strategies. We calculate the Kendall's Tau by predicting accuracy of all architectures in NAS-Bench-101 and comparing them with ground truths.

**Different evaluation numbers** We investigate the relationship between the number of evaluations performed on the validation dataset and the final Kendall's Tau. As shown in Fig.4, when the number of evaluations increases from 1 to 10, we can see a noticeable increase of Kendall's Tau. However, when the number of evaluations continues to grow, the increase of Kendall's Tau seems to be trivial but the training time increases a lot. Hence, we choose to evaluate on validation dataset 10 times in our self-evolution framework for all experiments.

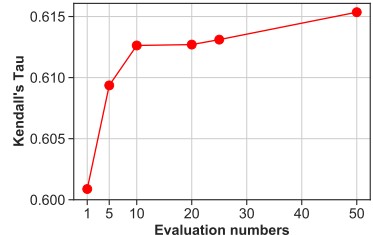

Figure 4: Kendall's Tau versus the number of validation evaluations.

**Ranking results over only good architectures** Although there are a large number of architectures in the search space, we usually only care about the top-performing ones. Therefore, we compared our method with the other two methods on NAS-Bench-101 [48] using Kendall's Tau as the metric, over different top portions of good architectures. Specifically, we only evaluate the good architectures whose ground truth accuracies rank in the top 10%, 20%, and 30%. The results are shown in Fig.5. When ranking over the top 10% architectures, NAO [30] and our TNASP get similar results and are better than Neural Predictor [43]. When ranking over more architectures i.e. top 20% and top 30%, TNASP shows a steady and better performance compared with NAO [30] and Neural Predictor [43].

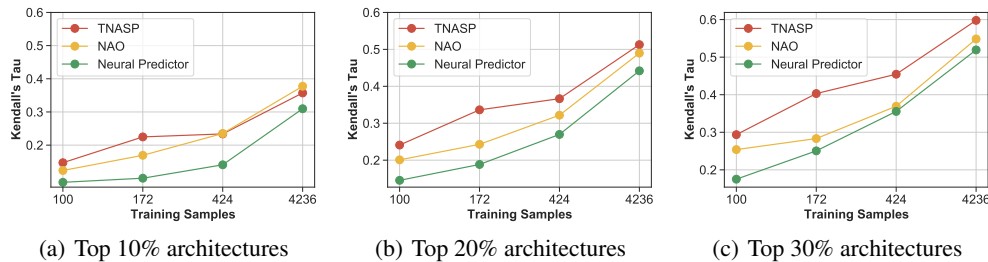

(a) Top 10% architectures    (b) Top 20% architectures    (c) Top 30% architectures

Figure 5: Ranking results over different top portions of good architectures.

## 5   Conclusion

In this paper, we propose a Transformer-based NAS performance predictor and utilize the linear transformation of the Laplacian matrix as the positional encoding. Our predictor has better encoding ability for spatial topology information, leading to state-of-the-art performance on several benchmarks. Moreover, we devise a general enough self-evolution framework to further improve our NAS predictor by fully utilizing the temporal information like historical evaluations during training. Unfortunately, we didn't explore how to involve complicated metrics, for example, non-differentiable metrics, as constraints in our framework. In the future, we will further explore how to choose more reasonable and effective constraints to improve the NAS predictors stably and efficiently.

# 6 Acknowledgements

This work is supported in part by the National Key R&D Program of China under Grant No. 2018AAA0102701, and in part by the National Natural Science Foundation of China under Grant No. 62176250. Moreover, we would like to thank Yu Hu for careful guidance; Longxing Yang, Peng Yao, Yuhang Jiao, and Wentao Zhu for insightful discussions.

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
