# TNASP: A Transformer-based NAS Predictor with a Self-evolution Framework - Supplementary Materials

**Shun Lu[1,2], Jixiang Li[3], Jianchao Tan[3], Sen Yang[3], Ji Liu[3]**

[1] Research Center for Intelligent Computing Systems, State Key Laboratory of Computer Architecture,
Institute of Computing Technology, Chinese Academy of Sciences
[2] University of Chinese Academy of Sciences
[3] Kuaishou Technology
lushun19s@ict.ac.cn, {lijixiang,jianchaotan,senyang,jiliu}@kuaishou.com

## A  Demonstration experiments

### A.1  Effects of two encoding branches in TNASP model design

We conduct an ablation study on NAS-Bench-101 [16] to further analyze the impact of the main components in our model. As shown in Tab.5, the absence of any encoding branch can lead to significant performance degradation, from which we can conclude that both positional encoding and operation encoding branches play an essential part in our TNASP design. Moreover, the operation encoding branch looks more important than the positional encoding branch. When both branches are adopted, our TNASP model achieves the best performance.

| POE | OPE | 100 (0.02%) all | 172 (0.04%) all | 424 (0.1%) 100 | 424 (0.1%) all | 4236 (1%) all |
|:---:|:---:|:---:|:---:|:---:|:---:|:---:|
|  | ✓ | 0.386 | 0.419 | 0.456 | 0.426 | 0.458 |
| ✓ |  | 0.229 | 0.314 | 0.366 | 0.303 | 0.376 |
| ✓ | ✓ | **0.600** | **0.669** | **0.752** | **0.705** | **0.820** |

Table 5: We analyze the effects of two encoding branches in TNASP model design under various data splits. OPE: operation encoding, corresponding to the top branch in Fig.1. POE: positional encoding, represents the bottom branch in Fig.1. When one of the branches is adopted, we shield the information from the other branch to analyze its effects on our TNASP model.

### A.2  Performance of replacing Transformer with GCN or MLP.

From our comparison and experiments, Transformer is necessary. For demonstration, we also have conducted additional experiments on NAS-Bench-101 [16] about replacing Transformer with GCN or MLP and results are shown in Tab. 6. When replacing our Transformer with GCN, we can get the models almost the same as the ones applied in NP (GCN) [14], which is obviously worse than our method. If replaced with MLP, since our regressor part is already MLP, that means to increase the depth of existing MLP, which however, would potentially have poor performance on such graph-structure data.

### A.3  Comparison with SemiNAS

As far as we know, SemiNAS [9] is the most similar work to ours, thus we provide detailed comparisons here. There are several essential differences between SemiNAS and TNASP. First, SemiNAS [9] directly utilize the NAO [10] model as their backbone implementation. On the contrary, we devise a

35th Conference on Neural Information Processing Systems (NeurIPS 2021).

| Backbone | 100 (0.02%) all | 172 (0.04%) all | 424 (0.1%) 100 | 424 (0.1%) all | 4236 (1%) all |
|---|---|---|---|---|---|
| MLP | 0.386 | 0.419 | 0.456 | 0.426 | 0.458 |
| GCN | 0.229 | 0.314 | 0.366 | 0.303 | 0.376 |
| Transformer | **0.600** | **0.669** | **0.752** | **0.705** | **0.820** |

Table 6: Replace Transformer with GCN or MLP on NAS-Bench-101.

novel and effective transformer based predictor associated with Laplacian matrix positional encoding strategy, which surpasses the NAO [10] model a lot on NAS-Bench-101 [16] and NAS-Bench-201 [5]. Second, SemiNAS [9] proposed to leverage the general heuristic semi-supervised framework to use testing set gradually augment the training set, which is usually hard to guarantee the optimization of the model in the right direction as the predicted data labels are inherently biased. They did propose a simple strategy to mitigate the bias, however, we propose a novel self-evolution framework to utilize the historical evaluation information over a validation set as constraints to supervise the model training, which can essentially guide the training towards the right direction gradually. Moreover, our method did not explicitly put the validation data into the training dataset as the pseudo label technique did.

To make a fair comparison, we conduct further experiments with SemiNAS [9] in Tab.7. When applied with the SemiNAS framework, we can see that several results only fluctuate around the original one (Compare three methods with/without mark "SM"). Yet, most ranking results get improved and all models achieve the best ranking results when applying our self-evolution framework (Compare three methods with/without mark "SE").

| Training Samples Validation Samples Test Samples | 100 (0.02%) 200 all | 172 (0.04%) 200 all | 424 (0.1%) 200 100 | 424 (0.1%) 200 all | 4236 (1%) 200 all |
|---|---|---|---|---|---|
| Neural Predictor[†] [14] | 0.391 | 0.545 | 0.710 | 0.679 | 0.769 |
| NAO[‡] [10] | 0.501 | 0.566 | 0.704 | 0.666 | 0.775 |
| TNASP | **0.600** | **0.669** | **0.752** | **0.705** | **0.820** |
| Neural Predictor[†] + SM | 0.392 | 0.547 | 0.713 | 0.680 | 0.770 |
| NAO[‡] + SM | 0.501 | 0.571 | 0.705 | 0.670 | 0.801 |
| TNASP + SM | **0.595** | **0.666** | **0.750** | **0.718** | **0.820** |
| Neural Predictor[†] + SE | 0.458 | 0.577 | 0.713 | 0.684 | 0.773 |
| NAO[‡] + SE | 0.564 | 0.624 | 0.732 | 0.680 | 0.787 |
| TNASP + SE | **0.613** | **0.671** | **0.754** | **0.722** | **0.820** |

Table 7: Comparison with SemiNAS [9]. We calculate the Kendall's Tau by predicting accuracy of architectures in NAS-Bench-101. [†]: re-implemented by ourselves. [‡]: implemented based on their released model. SM: SemiNAS [9] framework. SE: self-evolution framework.

## A.4 Comparison with BONAS

BONAS [12] adopted a GCN-based predictor to encode architectures and applied Bayesian optimization to search for architectures, while we introduced a Transformer-based predictor with novel positional encoding to encode spatial topology information and utilized the generalized self-evolution optimization framework to involve any available historical validation evaluation information to improve the model training.

Besides the difference between backbone models, in terms of how to use historical validation evaluation information, BONAS [12] trained a Bayesian regression model to make performance prediction, while our SE framework directly treats each historical validation evaluation information as each hard constraint during training and reformulate the whole constrained training problem as a minimax optimization problem, solved by gradient-based optimization method efficiently and

effectively, which we think is a pretty novel and different scheme comparing to BONAS. Our total design is to combine the spatial topology information encoding of input graph data and temporal evaluation information together to improve training stability and model generalization.

As for the performance, when evaluated on NAS-Bench-101, BONAS used 85% of the data for training while our method required at most 1% of the data. When searched in the DARTS search space, BONAS-A consumed 2.5 GPU days and achieved 97.31 test accuracy (BONAS-B/C/D required more GPU days) while our method only cost 0.3 GPU days and achieved $97.43 \pm 0.04$ test accuracy. Evidently, our method is novel, lightweight, efficient, and achieves SOTA performance.

### A.5 Experiments on ImageNet

To prove the effectiveness of our proposed NAS predictor, we also perform the architecture search on ImageNet dataset [7] in a MobileNet-like search space, which is composed of chain-like architectures and we apply the traditional positional encoding in NLP transformer for such sequential architectures. We search for the MobileNet block [11] with kernel sizes $\{3, 5, 7\}$ and expansion rates $\{3, 6\}$, and use the identity operation instead when there is no down-sampling. We first train a supernet with totally 21 layers for 120 epochs and then randomly select one thousand models to train our predictor. Finally, the evolutionary algorithm [3] is used to search for the superb models and we retrain them to get the final accuracy. All the training configures and the hyper-parameters follow the paper [6, 17].

The comparisons with other methods are summarized in Tab.8 and we visualize our searched architectures in Sec.C.2. We notice that TNASP-A has obtained the same or better accuracy than Proxyless (GPU) [1] and SPOS [6] with fewer parameters and FLOPs. When equipped with a little more parameters and FLOPs, TNPAS-C achieves the best performance than other predictor-based methods with the highest classification accuracy 75.8.

| Method | Params(M) | FLOPs(M) | Top-1(%) | Top-5(%) |
|---|---|---|---|---|
| FBNet-C [15] | 5.5 | 375 | 74.9 | 92.1 |
| Proxyless (GPU) [1] | 7.0 | 457 | 75.1 | 92.5 |
| SPOS [6] | 5.4 | 472 | 74.8 | - |
| RLNAS [17] | 5.3 | 473 | 75.6 | 92.6 |
| Neural Predictor [14] | 6.4 * | 536 * | $74.75 \pm 0.09$ | - |
| NAO [10] | 6.5 | 590 | 75.5 | 92.5 |
| TNASP-A | 5.0 | 433 | 75.1 | 92.3 |
| TNASP-B | 5.1 | 478 | 75.5 | 92.5 |
| TNASP-C | 5.3 | 497 | **75.8** | **92.7** |

Table 8: Comparison with other methods on ImageNet. *: We compute these information by their released model structure.

## B  Implementation details

**Predictor construction and training**   We set the embedding dimension of the operation encoding as 80 and build our predictor with 3 transformer encoder layers, each of which consists of 4 heads. The regressor module is composed of 2 fully-connected layers with hidden dimension 96. We train our predictor for 300 epochs with a small batch size of 10 on all search spaces. Adam optimizer is utilized with weight decay 1e-3 and we initialize the learning rate as 1e-4 with the cosine decay strategy.

**Applied with self-evolution framework and SemiNAS framework**   When applied with our self-evolution framework, we initialize the variable $\bar{y}$ using a normal distribution and sample 200 data as the validation dataset for evaluation. We further train the predictor using ADMM optimization for another 100 epochs (T in the equations of the main paper).

When applied with SemiNAS [9], we predict the accuracy of another 100 data and add them to the training set. We then use the newly generated training set to train the SemiNAS predictor for 50 epochs and we repeat the above iteration twice.

**Model evaluation** We retrain the searched cells in DARTS search space for 600 epochs with batch size 96, initial channels 36, and layers 20. The SGD optimizer is employed and we initialize the learning rate as 0.025 with a cosine decay strategy. We adopt the Cutout [4] to augment the training set and apply the Dropout [13] strategy in our training process. The auxiliary head is also utilized with a weight of 0.4. All of the training settings in DARTS search space follow the paper [8]. In the MobileNet-like search space, we retrain the searched architecture for 240 epochs with batch size 1024 on 8 NVIDIA V100 GPUs. Most of the training settings are kept the same as RLNAS [17]. We adopt the SGD optimizer and the initial learning rate is 0.5 with a cosine decay strategy. Label smoothing is applied with the weight of 0.1 and we clip the gradients larger than 5. We also use fixed AutoAugment policy [2] in our training.

We will release all the implementation code files, logs, and trained models once the paper is accepted.

## C Visualization

### C.1 Searched architectures in DARTS search space

We name three searched cells in DARTS search space as TNASP-a, TNASP-b, and TNASP-c. TNASP-a is the best searched cell and has been shown in our main text. TNASP-b and TNASP-c are shown in Fig.5. Normal cells are placed on the left side and reduction cells are visualized on the right side.

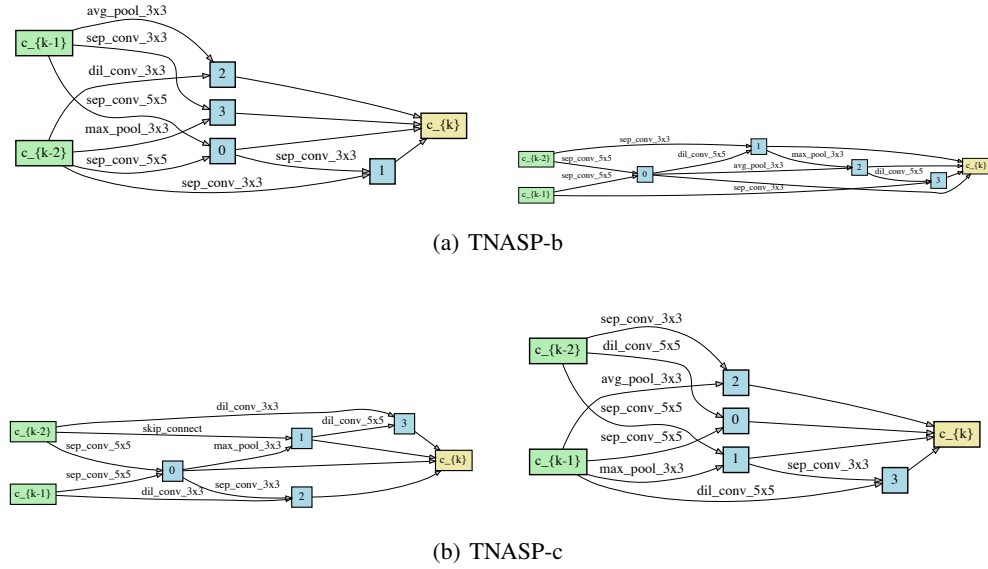

(a) TNASP-b

(b) TNASP-c

Figure 5: Other searched cells in DARTS search space. Left side are normal cells and right side are reduction cells.

### C.2 Searched architectures in MobileNet-like search space

We visualize our searched architectures TNASP-A, TNASP-B and TNASP-C in Fig.6.

## D Broader impact

Neural architecture search (NAS) has been a pretty active topic in recent years. This work mainly focuses on how to accurately predict the network performance efficiently during the neural architecture search, which can alleviate a large amount of computing cost and can help others construct more intelligent NAS-related systems. However, it still needs a few training data to train our predictor and requires expert experience to elaborately design the predictor and the search space for specific tasks.

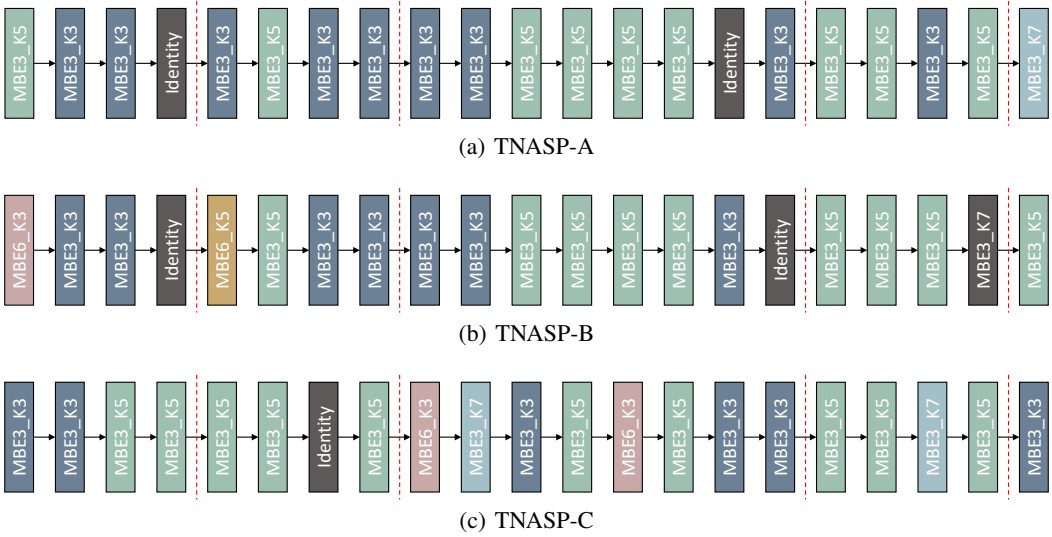

Figure 6: Our searched architectures in MobileNet-like search space.