# OpenReview forum: "TNASP: A Transformer-based NAS Predictor with a Self-evolution Framework"
_NeurIPS.cc/2021/Conference — NeurIPS 2021 Poster_

### Official Review · Reviewer_cAuS · 2021-07-12

**Rating:** 6
**Confidence:** 5

**Summary:**

This work leverages a transformer-based predictor network to approximate the accuracies of unseen architectures from a search space, for the purpose of NAS. The authors further proposed a ADMM-based method to accelerate the convergence of the predictor training. Experiments are conducted on NAS benchmarks and DARTS spaces.

**Limitations And Societal Impact:**

My main concern is the limited novelty of transformer-based NAS predictor, and the performance gain from self-evolution given the extra cost of hyper-parameter tuning.

**Main Review:**

1. I think the idea or graph or transformer-based predictor is reasonable. However, I do not think it is novel enough, given there are already many graph-based predictor works for NAS. I further named a few below that are not cited and compared by the authors.
2. The description from Eq. 8 to Eq. 13 is clear to me, but the flow char in Fig 2 is a bit confusing and less informative.
3. The self-evolution (SE) method further introduces extra hyper-parameters, the three $\eta$s in Eq.11~13. From Table 1 it seems the gain from SE is quite marginal. I am concerned about the cost of extra hyper-param tuning vs. this performance gain.


[1] Graph HyperNetworks for Neural Architecture Search

[2] D-VAE: A Variational Autoencoder for Directed Acyclic Graphs

[3] Neural Graph Embedding for Neural Architecture Search

[4] Fitting the Search Space of Weight-sharing NAS with Graph Convolutional Networks

**Time Spent Reviewing:**

1 hour

---

> ### Author Response · Authors · 2021-08-10
> **Response to Reviewer cAuS**
>
> **Q1. Doubts about the novelty.**
>
> A1. We have introduced a transformer-based NAS predictor with novel Laplacian-based positional encoding to achieve SOTA performance and proposed a model-agnostic self-evolution framework with good generalization to further improve the performance of our predictor, and our method achieved 2nd place in the recent international competition, demonstrating our whole designs are definitely novel and effective. In our paper, we have compared our method with recent state-of-the-art methods [5,6] and a GCN-based predictor[7], which obviously have worse performance than ours.
>
>
>
> Our novelty and contribution are two folds:
>
> * We demonstrate **the Transformer with the Laplacian matrix as the positional encoding is an effective way to encode graph-structure data**, which will inspire many works related to graph-structure data processing.
>
> * Our **model-agnostic self-evolution framework can directly treat each historical validation evaluation information as each hard constraint** during training and **reformulate the whole constrained training problem as a minimax optimization problem,** solved by gradient-based optimization method efficiently and effectively, which is pretty novel and make the trained model generalize well with limited training data.  Such SE framework can be adopted into many scenarios.
>
>
>
>
>
> **Q2. Comparison with four graph-based predictor papers.**
>
> A2. The related works[1,2,3,4] listed by the reviewer didn't release their code except for D-VAE[2], so it's hard to make a fair and full comparison due to the different experimental settings. We find **very recent works [5,6,7] cited in our paper didn't compare with them either,** but we are glad to cite and compare with them in our revised version.
>
> GHN[1] proposed to utilize Graph HyperNetwork (GNN+hypernetwork combination) to generate weights for sampled network architectures to help training the predictor. When proxylessly searched on the CIFAR-10 dataset, their method got **97.16±0.07** **test accuracy with** **0.84 GPU days and 5.7M parameters** while our method obtained **97.43±0.04 accuracy with only 0.3 GPU days and 3.6±0.1M parameters.** When transferring the searched model to the ImageNet dataset, their model only got **73.0 validation accuracy with 6.1M parameters** and our method achieved **75.2 validation accuracy with 5.3M parameters.** Our performance is much better in terms of these criteria. We will add this comparison to our revised version.
>
> D-VAE[2] adopted GNN to encode neural network architecture and applied an asynchronous message passing scheme to simulate the computation in a DAG. NAS task in this paper is just one application. As for the NAS experiment, D-VAE **used 90%** data for training predictor and use **only 10%** for testing, while our method required at most 1% data for training and still get good performance. For example, we find one common experiment performed on the CIFAR-10 dataset, the highest accuracy that D-VAE achieved is **94.80**, while our method obtained **97.48**, which is much better than theirs.
>
> NGE[3] simply apply the GCN to encode each node embedding and then concatenate embeddings together to represent the network architecture. When conducted searching on the same search space i.e. DARTS, our method achieved better results both on CIAFR-10 and ImageNet dataset
>
> (**ours: 97.43±0.04 and 75.2 versus NGE: 97.4 and 74.7**). We can not find other common experiments to compare.
>
> [4] claimed to train a GCN to alleviate the mismatch problem between assembled network layers and conducted architecture search on the MobileNet search space on the ImageNet dataset. They got **75.5** validation accuracy and further improved the result to **76.6** with an additional Squeeze-and-Excitation module. When aligning with their experimental settings, our best model TNASP-C achieved **75.8** validation accuracy and was further improved to **78.1** with an additional Squeeze-and-Excitation module, which is obviously better than [4].
>
> Thanks for pointing out these GCN-related works, and we already compared them in terms of methodology and performance comparison in this rebuttal. We can see that these related works' performance are all inferior to our method. We hope these will mitigate your concerns about our novelty and contribution.
>
>
>
>
>
> **Q3. Figure 2 is not very clear.**
>
> A3. We are sorry for the clarity problems in figure 2. We will improve it in a revised version later. Currently, we welcome the reviewer to read more details in equations (the symbols in equations correspond with the notations in Fig.2) and also our responses to Q4 and Q5 of Reviewer 8hPB, which may help make Fig.2 easier to understand.
>
>
>
> **Q4. The extra cost of hyper-parameters tuning of SE framework but with marginal gain.**
>
> A4.
>
> - Extra parameters tuning:
>
> For three parameters $\eta_{\theta}$, $\eta_{y}$ and $\eta_{\gamma}$ in equation 11 to 13, first $\eta_{\theta}$ is original model training learning rate, which is not the additional parameters. Only $\eta_{y}$ and $\eta_{\gamma}$ are introduced by the SE framework. During our experiments, we find our framework is robust to these two parameters. We tuned the parameters with a scale of 10 to find final values in paper quickly. By the way, our NAS predictor training is extremely fast (a few minutes) and thus making it very easy to tune these hyper-parameters from our SE framework, which is definitely worth the performance gain.
>
> - Marginal gain:
>
> Our SE framework can not only help improve our transformer-based predictor but also help improve the performance of NAO [9] and Neural Predictor [10], as shown at the bottom of Tab.1 and Tab.2, which demonstrate its generalization and effectiveness. The gain for our transformer-based predictor is not that marginal, since it is hard to improve a little over our already SOTA performance. And the gain for NAO [9] and Neural Predictor [10] is pretty obvious actually.
>
> We want to emphasize that our model-agnostic general self-evolution framework (solve a minimax optimization to leverage historical evaluation information) is novel enough and can fit for many situations like machine learning competitions, general model training, and so on, which will inspire other works in the community and this should be the **real GAIN** of our SE framework.
>
>
>
> **Reference**
>
> [1] Zhang C, Ren M, Urtasun R. Graph hypernetworks for neural architecture search[J]. arXiv preprint arXiv:1810.05749, 2018.
>
> [2] Zhang M, Jiang S, Cui Z, et al. D-vae: A variational autoencoder for directed acyclic graphs[J]. arXiv preprint arXiv:1904.11088, 2019.
>
> [3] Li W, Gong S, Zhu X. Neural graph embedding for neural architecture search[C]//Proceedings of the AAAI Conference on Artificial Intelligence. 2020, 34(04): 4707-4714.
>
> [4] Chen X, Xie L, Wu J, et al. Fitting the search space of weight-sharing nas with graph convolutional networks[J]. arXiv preprint arXiv:2004.08423, 2020.
>
> [5] Chen Y, Guo Y, Chen Q, et al. Contrastive Neural Architecture Search with Neural Architecture Comparators[C]//Proceedings of the IEEE/CVF Conference on Computer Vision and Pattern Recognition. 2021: 9502-9511.
>
> [6] Xu Y, Wang Y, Han K, et al. ReNAS: Relativistic evaluation of neural architecture search[C]//Proceedings of the IEEE/CVF Conference on Computer Vision and Pattern Recognition. 2021: 4411-4420.
>
> [7] Wen W, Liu H, Chen Y, et al. Neural predictor for neural architecture search[C]//European Conference on Computer Vision. Springer, Cham, 2020: 660-676.
>
> [8] Shi H, Pi R, Xu H, et al. Bridging the gap between sample-based and one-shot neural architecture search with bonas[J]. arXiv preprint arXiv:1911.09336, 2019.
>
> [9] Luo R, Tian F, Qin T, et al. Neural architecture optimization[J]. arXiv preprint arXiv:1808.07233, 2018.
>
> [10] Wen W, Liu H, Chen Y, et al. Neural predictor for neural architecture search[C]//European Conference on Computer Vision. Springer, Cham, 2020: 660-676.

---

> > ### Comment · Reviewer_cAuS · 2021-08-20
> > **Thanks for the response**
> >
> > I appreciate the author's careful response and effort. After reading I decide to raise the score to 6, mainly due to the detailed comparison against other graph-based NAS predictors. However, I still believe the transformer-based NAS predictor is a reasonable choice, but not a significant contribution to the NAS community.

---

> > > ### Author Response · Authors · 2021-08-20
> > > **Great thanks to the reviewer's comments.**
> > >
> > > We are really grateful to the reviewer for the thoughtful comments and for improving the score. We will add detailed comparisons with these four graph-based NAS predictor methods in our revised version.
> > >
> > > In our paper, our proposed transformer-based NAS predictor is proved to be effective and efficient in different NAS tasks, due to **significant performance improvements and search cost savings** compared with other methods, as shown in our paper and our response to other reviewers.
> > >
> > > However, this is only one of our contributions. Moreover, we have also explored various positional encoding methods (including randomly generated ones, adjacency matrix+MLP, eigenvectors of Laplacian matrix, and our proposed laplacian matrix+MLP) for our transformer-based NAS predictor as shown in Tab.4 and our response to other reviewers. **The novelty is obvious and the explainability is straightforward.**
> > >
> > > The above two contributions may still be in the same scope as other accepted top-tier conference papers in the NAS predictor field. **In this scope, we have already obtained SOTA performance.  Besides this scope, we further proposed a generic and model-agnostic self-evolution framework** to improve the performance of NAS predictor by utilizing previous evaluation information as constraints, which we think can be applied to many application scenarios. Therefore, we believe that **with all of these three contributions together, our work is definitely able to provide significant insights for the NAS community.**
> > >
> > > Thank you again for your time. We hope our new response can mitigate your concerns over our contributions.

---

### Official Review · Reviewer_fP1R · 2021-07-16

**Rating:** 5
**Confidence:** 4

**Summary:**

This paper proposes a transformer-based NAS performance predictor to better encoding the spatial topology information and introduces a self-evolution framework to further improve the performance of the predictor. The proposed method achieves SOTA results on three benchmarks.

**Limitations And Societal Impact:**

Yes

**Main Review:**

Pros:

(1)	A transformer-based predictor and a self-evolution framework are proposed, and the experiments on different benchmarks verify the effectiveness of the proposed method.

(2)	The self-evolution framework makes full use of the dataset by leveraging the validation loss. When the size of the training dataset is small, it improves the generalization of the model.

Cons:

(1)	Though the author said that the self-evolution framework is practical in machine learning competitions, the test dataset in competitions is usually not provided to users to store the predictions for validation loss, which limits the usage of the proposed framework.

(2)	The detailed experiment setting of the transformer-based predictor is not reported.

(3)	The author said the core factor of the proposed predictor is the better encoding ability for spatial topology information. However, it is not convincing that Laplacian matrix-based positional encoding does represent topology information. As a comparison, what is the performance of transformer-based predictor associated with random initialized positional embedding? Similarly, the Laplacian matrix is also used by GCN-based predictor which strictly limits the direction of information aggregation by Laplacian matrix. It would be better to provide the experiment results of GCN-based methods.

(4)	The main idea of self-evolution framework is similar to the pseudo label technique that SemiNAS adopts. What is the performance of SemiNAS on these benchmarks?

(5)	Why is lambda in Eq. (10) must be greater than or equal to 0?

(6)	How does the proportion of training set and test set affect the performance of self-evolution framework?

(7)   There are some recent/concurrent works on NAS missing in the related works, such as "CATE: Computation-aware Neural Architecture Encoding with Transformers", "Vision Transformer Architecture Search" and "AutoFormer: Searching Transformers for Visual Recognition". It is suggested to add them into comparison and discussion, especially for CATE. Recent papers on NeurIPS'20 are also not included in the comparison.

Typos:
(1)	In line 127, acrylic may be acyclic.
(2)	In line 128, N may denote the number of nodes, not the connections (edges).
(3)	In line 4 of Algorithm 1, the new constraint may be added into Eq. (8), not Eq. (10)


**Time Spent Reviewing:**

5 hours

---

> ### Author Response · Authors · 2021-08-10
> **Response to Reviewer fP1R**
>
> We thank the reviewer for insightful thoughts and good suggestions.
>
> **Q1. Practical usage of the self-evolution framework.**
>
> A1. In our recent competition, the test dataset is available for us to compute predictions first and then submit predictions to the system to get evaluation information, thus our self-evolution framework just hits the spot and we find it truly helps further improve model performance in the competition.
>
>
>
> When the test dataset is inaccessible, we can also **split a small portion of training data as the validation dataset to obtain validation evaluation results as hard constraints for the SE framework,** which is also just the case in our NAS benchmarks experiments in the paper (we sampled some data as the validation dataset to obtain constraints to help the training of our transformer-based predictor). And for this case, we find the trained model can generalize well and perform well on a large portion of the unseen test dataset as shown in our paper, demonstrating the effectiveness of the self-evaluation framework.  Therefore, our framework is generic in practical whether or not the test dataset is available.
>
>
>
> **Q2. Experimental details.**
>
> A2. Due to page limitation, we already put implementation details and experimental settings of our Transformer-based predictors into Sec.B of the supplementary material, sorry for the inconvenience. And we **will release all the code and models upon acceptance.**
>
>
>
> **Q3. Doubts about the Laplacian-matrix-based positional encoding and comparison with GCN models.**
>
> A3. When using randomly initialized positional encoding for our transformer-based predictor, we got Kendall's Tau (0.374, 0.366, 0.469, 0.362, 0.365), even worse than the one that does not use positional encoding at all (shown in Tab.5) on NAS-Bench-101, with the same 5 data splits. It is definitely worse than our proposed Laplacian-matrix-based positional encoding.  Because the introduction of additional random noise into operation feature representation will make the training unstable and hard to converge.
>
>
>
> For the GCN-based predictor, adopting the Laplacian matrix into GCN actually got inferior results than adopting adjacency matrix into GCN, according to our previous experiments, but **it's not the case for our Transformer-based predictor**, where we got best results under different data splits if leveraging the (Laplacian matrix +MLP) to be positional encoding, as shown in Tab.4.
>
> **Q4. Comparison with SemiNAS and its pseudo label technique.**
>
> A4. The pseudo label technique is to every time put a small portion of validation data into the training dataset during the loop of training, making the model performs better and better gradually.
>
>
>
> We want to emphasize that our SE framework directly treats each historical validation evaluation information as each hard constraint during training and reformulate the whole constrained training problem as a minimax optimization problem, solved by gradient-based optimization method efficiently and effectively, which is pretty novel and make the trained model generalize well with limited training data. **We did not explicitly put the validation data into the training dataset as the pseudo label technique did.**
>
>
>
> Due to page limitations, we already placed the performance comparisons with SemiNAS [1] in A.3 of our supplementary material. When applied with the same number of training and validation data for a fair comparison, we deploy our SE framework and pseudo label technique onto three different kinds of predictor models. We found that our self-evolution framework can robustly achieve better performance than the pseudo label technique from SemiNAS [1] on all three predictor models.
>
>
>
> **Q5. Question about lambda in Eq. (10).**
>
> A5. $\lambda$ is the Lagrange multiplier. We use the Lagrange multiplier method to reformulate a hard-constrained optimization problem (Eq.8) into a minimax optimization problem (Eq.10).  The multiplier for our equality constraint does NOT need to be greater than or equal to zero. Thanks for pointing out this error due to our carelessness, we will fix it in revised version.
>
>
>
> **Q6. Influence of different splits on train and validation dataset on the SE framework.**
>
> A6. As shown in Tab.1 and Tab.2, when the number of validation samples is greater than training samples, the predictor can get a larger performance improvement by using the SE-framework. When the number of training samples is much larger than validation samples, the predictor only got little performance gains. This phenomenon happens **not only on our transformer-based predictor but also on NAO [2] and Neural Predictor [3]**, as shown at the bottom of Tab.1 and Tab.2.
>
>
>
> **Q7. Related works**
>
> A7. CATE[4] was announced to be accepted by ICML 2021 at the end of May, which was close to the NeurIPS paper deadline so we missed it, but we are glad to add it as the related work.  We compare with it in terms of model complexity, training style, model performance below:
>
>
>
> - CATE[4] applied more transformer encoder blocks than ours, resulting in more parameters (encoding module from their released code: **CATE: 1.61M vs TNASP: 0.496M** on NAS-Bench-101) and required far more training data than ours to get comparable performance (on NAS-Bench-101, it used 95% of the data for training and tested on 5% held-out test sets, while our method required at most 1% of the data and tested on the whole dataset).
> - Moreover, **CATE[4] is a two-stage method** and demands selecting similar architecture pairs to pre-train the encoding module first, which is computationally intensive as claimed by themselves, however, **our method is end-to-end** and does not require selecting paired training data at all during training.
> - To get a more fair comparison, we notice that when proxylessly searched on DARTS search space with a small budget, our method (**0.3 GPU days**) only consumed less than a tenth of the search cost than CATE (**3.3 GPU days**) but achieved similar performance (**ours:97.43±0.04 vs CATE:97.45±0.08**), which can demonstrate the superiority of our method.
>
>
>
> ViTAS[5] and AutoFormer[6] mainly focused on searching for optimal configurations for the Transformer architecture itself, while our work utilized the Transformer architecture as a predictor to predict the performance of input model architecture. **We are in orthogonal domains.**  Moreover, these two works are arXiv papers after June and **invisible to us before our NeurIPS submission deadline**, but we are glad to add them as related works.
>
>
>
> Thanks for pointing out these related works and several typos, we will fix them in the revised paper.
>
>
>
> **Reference:**
>
> [1] Luo R, Tan X, Wang R, et al. Semi-supervised neural architecture search[J]. arXiv preprint arXiv:2002.10389, 2020.
>
> [2] Luo R, Tian F, Qin T, et al. Neural architecture optimization[J]. arXiv preprint arXiv:1808.07233, 2018.
>
> [3] Wen W, Liu H, Chen Y, et al. Neural predictor for neural architecture search[C]//European Conference on Computer Vision. Springer, Cham, 2020: 660-676.
>
> [4] Yan S, Song K, Liu F, et al. CATE: Computation-aware Neural Architecture Encoding with Transformers[J]. arXiv preprint arXiv:2102.07108, 2021.
>
> [5] Su X, You S, Xie J, et al. Vision Transformer Architecture Search[J]. arXiv preprint arXiv:2106.13700, 2021.
>
> [6] Chen M, Peng H, Fu J, et al. AutoFormer: Searching Transformers for Visual Recognition[J]. arXiv preprint arXiv:2107.00651, 2021.

---

> > ### Comment · Reviewer_fP1R · 2021-09-01
> > **Thanks for the response**
> >
> > I appreciate the authors taking the time, attempting to address the comments through proving more details. After reading the authors' response, I am still hesitant to recommend acceptance of the current submission. I do see the merit of the method proposed and believe it would benefit tremendously from improved presentation and quality in a future submission. Thanks.

---

> > > ### Author Response · Authors · 2021-09-01
> > > **Great thanks to the reviewer's comments.**
> > >
> > > We thank the reviewer's comments and appreciate that the merit of our method is well recognized by the reviewer.
> > > The improvements of paper presentations and reformulations can be made straightforwardly by incorporating our responses and all reviewers' comments into our original paper, which can be done in a short time.
> > >
> > > We still hope the reviewer can consider the merits of our method and its inspirations to the community:  obviously better predictor performance than SOTA on all benchmark experiments;  a transformer model with a novel design of positional encoding to process graph structure data effectively, which is also explainable;  a general self-evolution framework to further improve generalizability by using constrained optimizations formulation, which can be applied in many scenarios.
> > >
> > > If the reviewer **still has some specific concerns**, we are glad to provide more discussions and explanations to mitigate them further. Thanks.

---

### Official Review · Reviewer_bjoz · 2021-07-19

**Rating:** 5
**Confidence:** 5

**Summary:**

This paper proposed a transformer-based NAS performance predictor, accompanying with a Laplacian matrix based positional encoding to represent spatial topology information and a self-evolution mechanism to take advantage of the temporal information such as historical evaluations.

**Limitations And Societal Impact:**

Please justify the weakness to improve the paper.


**Main Review:**

This paper is well-organized with solid experiments in both the main paper and supplementary material.

Strength:
+ Using Laplacian matrix  to model spatial topology information is somewhat novel
+ Good performance on NAS-Bench 101, 201

Weakness:
There are some issues needed to be addressed before accepted.
- The claimed contributions (positional embedding and self-evolution) are closely related to BONAS from NIPS 2020. What is the difference and advantage over BONAS?
- Is transformer necessary? How about replacing transformer with GCN or MLP?
- In the ImageNet experiment, the performance gap between TNASP-A,B,C is big while the flops difference is small. What is the reason?  What is the variance between different runs?
- Potential double blind violation: "Our proposed method ranked 2nd among all teams in one recent international NAS challenge." This is a near miss. I will warn the author not to play this trick next time, if not deck rejected this time.

**Time Spent Reviewing:**

2

---

> ### Author Response · Authors · 2021-08-10
> **Response to Reviewer bjoz**
>
> Thank you for your instructive comments and suggestions.
>
> **Q1. Difference and advantage over BONAS.**
>
> A1. BONAS [1] adopted a GCN-based predictor to encode architectures and applied Bayesian optimization to search for architectures, while we introduced a Transformer-based predictor with novel positional encoding to encode spatial topology information and utilized the generalized self-evolution optimization framework to involve any available historical validation evaluation information to improve the model training.
>
> Besides the difference between backbone models, in terms of how to use historical validation evaluation information, BONAS [1] trained a Bayesian regression model to make performance prediction, while our SE framework directly treats each historical validation evaluation information as each hard constraint during training and reformulate the whole constrained training problem as a minimax optimization problem, solved by gradient-based optimization method efficiently and effectively, which we think is a pretty novel and different scheme comparing to BONAS [1].
>
> Our total design is to combine the spatial topology information encoding of input graph data and temporal evaluation information together to improve training stability and model generalization.
>
> Experiments comparison: when evaluated on NAS-Bench-101, BONAS [1] used **85%** of the data for training while our method required **at most 1%** of the data. When searched in the DARTS search space, BONAS-A consumed **2.5 GPU days** and achieved **97.31** test accuracy (BONAS-B/C/D required more GPU days) while our method only cost **0.3 GPU days** and achieved **97.43±0.04** test accuracy.  Our method is novel, lightweight, efficient, and achieves SOTA performance.  Thanks for pointing out this related work and we will add this comparison in our revised paper.
>
>
>
> **Q2. Performance of replacing Transformer with GCN or MLP.**
>
> A2. From our comparison and experiments, Transformer is necessary. When replacing our Transformer with GCN or LSTM, we can get the models almost the same as the ones applied in Neural Predictor (GCN) [2] and NAO (LSTM) [3], which both are obviously worse than our method as shown in Tab.1/2/7.  If replacing with MLP,  since our regressor part is already MLP, that means to increase the depth of existing MLP, which however, would potentially have poor performance on such graph-structure data.  For demonstration, we also have conducted additional experiments on NAS-Bench-101 about replacing Transformer with GCN or MLP. When replacing transformer with GCN, we obtained 0.286, 0.413, 0.682, 0.579, 0.747 Kendall's Tau under 5 data splits, which are obviously worse than ours. When replacing transformer with MLP, we got even worse performance: 0.268, 0.451, 0.566, 0.562, 0.615.  The data splits are the same as Tab.1.
>
>
>
> **Q3. Large performance gap in different runs, but with small FLOPs difference, and what's the variance between different runs.**
>
> A3. Although there are small FLOPs differences between different runs, however, these FLOPs may correspond to pretty different final model structures and weights, thus the performance may be pretty different. Similar to our case, as shown in Tab.7, the NAO [3] model only has 54M more FLOPs than Neural Predictor [2] models but can achieve 0.75 higher accuracies.  Our searched models are visualized in Fig.7 and we will release all the pre-trained weights upon acceptance.
>
> By the way, the data of Neural Predictor [2] in Tab.7 is chosen from the highest value from Tab.2 in this paper's supplementary material and the variance is caused by 10 random splits of the dataset in the cross-validation. More details can be found in the supplementary material of Neural Predictor[2]. We fixed all the seeds and adopted the official validation dataset as most previous works did, thus our model didn't have this kind of variance between different runs.
>
>
>
> **Q4. Potential double blind violation.**
>
> A4. Before submitting our paper, we **have emailed the NeurIPS official staff to consult is it possible to properly express the good competition results** in our paper to demonstrate our framework can be useful, and they think it is ok to show such demonstration, and suggested we stated it in the way we used in the paper.
>
> [Here is our email on May 25th, 2021] :
>
> "Dear Chairs, Sorry to bother you. I have a question about the policy of mentioning the international challenge details in paper submission (the challenge finished before the conference deadline). Is it ok to mention in paper like "Our method ranked 1st in recent international challenge" or similar words, without mentioning the name of this challenge?"
>
> [Here is the response from NeurIPS PCs on May 26th, 2021] :
>
> "It should be fine but please make it clear that you are not specifying the name of the challenge to preserve author anonymity, and all details will be added in the final version if the paper is accepted."
>
> We follow the instructions from this official email and sorry for the misunderstanding.
>
>
>
> **Reference:**
>
> [1] Shi H, Pi R, Xu H, et al. Bridging the gap between sample-based and one-shot neural architecture search with bonas[J]. arXiv preprint arXiv:1911.09336, 2019.
>
> [2] Wen W, Liu H, Chen Y, et al. Neural predictor for neural architecture search[C]//European Conference on Computer Vision. Springer, Cham, 2020: 660-676.
>
> [3] Luo R, Tian F, Qin T, et al. Neural architecture optimization[J]. arXiv preprint arXiv:1808.07233, 2018.

---

### Official Review · Reviewer_8hPB · 2021-07-19

**Rating:** 6
**Confidence:** 5

**Summary:**

This paper proposed a predictor based on the Transformer architecture to predict the performance of neural architectures in NAS.
It proposed to use Laplacian matrix based positional encoding strategy to better represent the topology information to model the neural architectures.
It also proposed a model-agnostic self-evolution framework that can fully utilize temporal information as guidance.

**Ethics Review Area:**

["I don’t know"]

**Main Review:**

Predictor plays an important role in NAS and predictor-based NAS is straight and simple for deploying in practice.
Besides previous LSTM/GNN based predictors, the authors proposed to use the advanced Transformer architecture for modeling the neural network architectures to predict the performance (e.g., accuracy).
Given the experiment results and the announced "... 2nd among all teams in one recent international NAS challenge", the methods demonstrate its effectiveness.

Strengths:
1. The proposed positional encoding seems to bring the benefit, compared to [1] where the authors showed that directly applying Transformer to the task gets bad performance. Does the gain come from the positional encoding?

Weaknesses and questions:
1. There lacks an introduction of Laplacian matrix but directly uses it. A paper should be self-contained.
2. The motivation is not strong. The authors stated that "... the transformer architecture ... outperformed many SOTA models ... motivates us". This sounds like "A tool is powerful, then I try the tool on my task, and it works well! Then I publish a paper". However, it lacks of analysis why Transformer works well on this task, which would bring more insights to the community.
3. Section 3.3 needs to be polished more on writing.\
1 In Eqn. (8), The e^{(t)} is proposed without further explanation. What is it? Why is it needed? What is the motivation of proposing it?\
2 In Eqn. (8), are f_\theta(v_j) and \hat{y}_j^{(t)} the same thing since they both are the predictions of v_j by the predictor.\
3 In line 177-178, if the y_j^{true} is "user-unkonwn", then how do you compute e^{(t)} in Enq. (8)?\
4 Why this SE framework can help to improve, how does it help? Similar to 2, please DO NOT just show me what you have done and achieved, but also show me why and how you manage to do these.

I would consider increasing the rating based on the authors' response.

Reference:
[1] Luo, et al. "Neural architecture search with gbdt." arXiv preprint arXiv:2007.04785 (2020). https://arxiv.org/abs/2007.04785

**Time Spent Reviewing:**

6

---

> ### Author Response · Authors · 2021-08-10
> **Response to Reviewer 8hPB**
>
> We sincerely thank reviewer 8hPB for insightful comments and great suggestions.
>
> **Q1. Compared to [1], does the gain come from the positional encoding?**
>
> A1. As shown in Tab.5 of our supplementary material, both our operation embedding module and our positional encoding design are essential for our Transformer-based predictor. [1] is an arXiv paper and there are very few details about Transformer models in their ablation study. Importantly, we are unaware of their pre-processing techniques of input graph data. Therefore, it's hard to say why their Transformer-based predictor got bad performance. Our positional encoding design does play a major role in the good performance. Moreover, we have verified in Tab.4 that even directly applying the (adjacency matrix + MLP) as the positional encoding can obtain better performance than most previous NAS predictor works, which however, is inferior to our (Laplacian matrix + MLP) option.
>
> **Q2. Lack an introduction of the Laplacian matrix.**
>
> A2. We will add an introduction for it in revision.  Laplacian matrix is usually computed by the difference between degree matrix and adjacency matrix, commonly used in graph-related processing, which encodes the relationship of neighborhood nodes in a graph, the connectivity of a graph, and so on. We find it has the potential to represent topology information of graph structure data. Thus we choose to use (Laplacian matrix + MLP) as the positional encoding for our Transformer-based predictor. Furthermore, we test other methods for positional encoding, for example, using eigenvectors of Laplacian matrix as positional encoding, using (adjacency matrix + MLP) as positional encoding. However, all of these are worse than our current way.
>
> **Q3. Motivation is not strong and why Transformer works well.**
>
> A3. Our motivation is to further push the upper boundary of the performance of previously proposed predictors with limited training data. As we mentioned in the abstract and introduction, current predictors did not capture the good feature representation of input graph structure data due to their bad encoding design on topology information and operations embedding features. To overcome this problem, we choose to take advantage of the powerful Transformer structure and further modify its positional encoding way to successfully achieve SOTA performance.
>
> We have analyzed the effectiveness of our Transformer-based predictor at Line 46-52 in the Introduction Section and Line 156-160 of Sec.3.2. Specifically, we want to emphasize the reasons here again.There are several reasons that our proposed Transformer-based framework works well as a NAS predictor:
>
> First, the attention module can help explore better feature representations from the graph structure data.
>
> Second, the multi-head mechanism can further help encode the different subspace information at different positions from the graph structure data, as also claimed by the original Transformer paper [2].
>
> Third, the Laplacian-matrix-based positional encoding method also fits well to find topology information on the graph as shown in Tab.4.
>
> In summary, we demonstrate that Transformer is an effective method to extract feature representation from discrete architecture graphs, and also has superb generalization abilities for processing unseen test data, as shown in our experiments.
>
> **Q4. Doubts about writing in Section 3.3.**
>
> A4.
> 1. $e^{(t)}$ represents the evaluation error (We choose to use common MSE function in equation 9) between predictions  $\hat{y}_j^{(t)}$  and ground truth  $y_j^{true}$ at t-th evaluation, on validation dataset V. We use each  $e^{(t)}$  as each hard constraint during training in our SE framework (equation 8).  The constraints mean that we try to find a proxy variable $\bar{y}_j$ (proxy variable for unknown ground truth),  whose evaluation error with respect to the prediction $\hat{y}_j^{(t)}$ **should be equal to $e^{(t)}$**. With such constraints, our framework can make full use of previous evaluation information to guide the training gradually.
>
> 2. **$f_\theta(v_j)$ and $\hat{y}_j^{(t)}$ are not exactly the same thing.** $f_\theta(v_j)$ is current forward pass inference results of predictor $f_\theta(x)$ over validation data $v_j$, which is used in back-propagation step to update parameters $\theta$ at current step. $\hat{y}_j^{(t)}$ is the corresponding PREVIOUS predictions at time step $t=1, 2, 3, ..., n$. Thus we call our framework as "self-evolution", or supervised by historical evaluations temporarily.
>
> 3. When we say that $y_j^{true}$ is "user-unknown", **it means different in different situations**:
>
>    * If in the NAS competition, at t-th submission, we submit our predictions $\hat{y}_j^{(t)}$ to system judger, and the system will compute the $e^{(t)}$ at the backend privately and return only one value $e^{(t)}$  to us, and thus we call $y_j^{true}$ is user-unknown.  Our self-evolution system can make use of only these $e^{(t)}$ for $t=1, 2, 3, ..., n$ to incrementally improve our predictor.
>
>    * If in offline benchmark experiments, we use evaluation error on validation set V as $e^{(t)}$ to guide the training using our SE framework.  In this case, we did know the ground truth on this validation set V, but we did not make use of this ground truth directly in the SE framework. We just mimic the aforementioned international competition system to compute $e^{(t)}$ as pre-processing privately. Thus the ground truth is also "unknown".
>
> **Q5. Why the Self-Evolution (SE) framework works well.**
>
> A5. SE framework can make full use of any available information (either the competition system historical submission feedback or historical validation errors during model training) to guide the predictor model training to avoid over-fitting, thus generalized well on the test dataset.  **Our SE framework directly treats each historical validation evaluation information as each hard constraint during training and reformulate the whole constrained training problem as a minimax optimization problem, solved by gradient based optimization method efficiently and effectively.**
> Specifically, follow the notations in Answer 4,  $\bar{y}_j^{(t)}$ is an auxiliary variable in the SE ADMM framework to be the proxy variable for the unknown ground truth. We use this proxy variable to regularize the current time prediction $f_\theta(v_j)$ and previous prediction $\hat{y}_j^{(t)}$. Every time we are collecting previous evaluation error $e^{(t)}$  for $t=1, 2, 3, ..., n$ to improve system gradually. When the system converges, $\bar{y}_j^{(t)}$, $f_\theta(v_j)$ and $\hat{y}_j^{(t)}$ should be close enough, and also be close to unknown ground truth. Therefore, our SE framework can make predictors generalize well even with limited training data, thus we can get good predictor performance on the test dataset.
>
> **Summary:**
>
> We will improve the clarity of equations and symbols, add missed references, and emphasize more on why the proposed method can work well in the revised paper.
>
>
> **Reference:**
>
> [1] Luo, et al. "Neural architecture search with gbdt." arXiv preprint arXiv:2007.04785 (2020).
>
> [2] Vaswani A, Shazeer N, Parmar N, et al. Attention is all you need Advances in neural information processing systems. 2017: 5998-6008.

---

### Decision · Program_Chairs · 2021-09-27

**Decision:**

Accept (Poster)

**Comment:**

The authors apply transformer with Laplacian matrix positional encoding as a predictor in predictor based neural architecture search. They also introduce a self-evolution framework to incorporate new data points to improve the performance. They test the model on standard NAS and DARTS benchmarks (collected set of architectures with their performances) and achieve some of the best results.

Overall the paper is sound. The main issue from the reviewers was if the paper is sufficiently novel to be accepted to neurips and also partially the small size of the NAS and DARTS benchmarks as an evaluation for the neural architecture search. There were also a lot of question to relations to other works and reference that were missing in the paper, however the authors have clarified a lot of the differences from the previous works in the rebuttal.